# Influence of Geometric and Material Uncertainties on the Behavior of Monostable and Bistable Electromagnetic Energy Harvesters

**DOI:** 10.3390/s26010253

**Published:** 2025-12-31

**Authors:** Petr Sosna, Zdeněk Hadaš

**Affiliations:** Faculty of Mechanical Engineering, Brno University of Technology, 616 69 Brno, Czech Republic; hadas@fme.vutbr.cz

**Keywords:** kinetic energy harvesting, nonlinearity, electromagnetic harvester, uncertainty analysis, bistability

## Abstract

Uncertainties in geometry, material properties, and excitation forces critically influence the performance of nonlinear electromagnetic vibration energy harvesters, which are promising power sources for wireless sensor networks in industrial environments. These nonlinear harvesters rely on tunable magnetic stiffness to achieve broadband operation, but their strong nonlinear coupling makes them highly sensitive to small parameter deviations. This study investigates how geometric tolerances, variability of magnetic material properties, and excitation irregularities affect the dynamic response and harvested output power of electromagnetic vibration energy harvesters. Nonlinear magnetic restoring forces were obtained using Finite Element Method Magnetics simulations and implemented in a one-degree-of-freedom model for numerical analysis. The results show that deviations as small as ±0.1 mm in geometry or ±5% in magnetic coercivity can shift the system between monostable, bistable, and chaotic regimes, which could dramatically change wireless sensor operation. Controlled asymmetry of design and impulsive excitation were found to facilitate high-energy orbits, enhancing stability and energy conversion. These findings demonstrate that understanding and managing uncertainty amplification across geometric, material, and excitation domains is essential for reproducible and reliable operation, supporting the design of robust nonlinear electromagnetic harvesters for industrial applications of wireless sensor networks.

## 1. Introduction

The growing demand for autonomous monitoring in industrial systems, such as aerospace structures [1], railway systems [2,3], and civil infrastructure [4,5], has intensified interest in vibration energy harvesting (VEH) as a sustainable power source for wireless sensor networks (WSNs). By converting ambient mechanical vibrations into electricity, VEH systems eliminate the need for frequent battery replacement and enable long-term, maintenance-free operation in hard-to-access environments [6]. Among various transduction mechanisms, piezoelectric and electromagnetic approaches remain the most promising for practical deployment due to their compactness, scalability, and compatibility with typical industrial vibration frequencies.

Piezoelectric devices can generate high voltages and are structurally compact [7,8]; however, they suffer from high damping [9] and material fatigue under long-term cyclic loading [10]. Electromagnetic (EM) harvesters, in contrast, offer robust mechanical performance and higher efficiency at low frequencies typical of large-scale machinery [11,12]. The main limitation of conventional, linear VEH systems is their narrow operational bandwidth, which restricts efficiency under variable excitation. To overcome this, researchers have explored nonlinear [13] and tunable [14] configurations that broaden the response spectrum and enhance adaptability.

While hybrid [15,16,17,18] and tristable [19,20] configurations have been proposed, they remain valuable primarily from a theoretical perspective for exploring complex nonlinear dynamics [21]. However, their increased structural and control complexity often outweighs the potential performance gains in practical implementations. A notable exception among hybrid configurations are systems in which one subsystem is dedicated solely to tuning rather than dual-mode energy extraction. For example, the electrostatic–piezoelectric MEMS design proposed in [22] employs the piezoelectric element for harvesting while the electrostatic actuator serves exclusively as a resonance-tuning mechanism, thereby avoiding the added complexity typically associated with hybrid harvesters. Similarly, chaotic designs, once considered attractive for energy amplification, have proven unreliable and unpredictable for industrial deployment [23,24,25].

Nonlinear electromagnetic energy harvesters with adjustable magnetic stiffness [26] have shown particular promise. By varying the spacing between repelling permanent magnets, the restoring force can be tuned to change the harvester’s effective stiffness and to achieve either monostable or bistable behavior, enabling the system to adapt to changing vibration conditions. Bistable designs, in particular, can reach high-amplitude “high-orbit” oscillations that significantly increase power output [27]. However, this beneficial regime coexists with low-energy and chaotic attractors, making the nonlinear energy harvesting system response highly sensitive to parameter variations.

The primary experimental results of the linear laboratory sample [28] and industrial prototype [29] (see Figure 1A) have been published, and such railway WSN applications expect mass deployment of harvesters. Both studies addressed trackside kinetic electromagnetic energy harvesters designed for powering wireless sensor nodes, demonstrating their feasibility in railway infrastructure operation based on WSNs. Although these systems were essentially linear, the concept presented in [28] offers a straightforward path toward nonlinear and tunable behavior by the addition of magnetic elements. This opens a new research question regarding the transition from linear laboratory devices to nonlinear, universally adaptable harvesters suitable for robust industrial integration.

A key remaining challenge is the nonlinear energy harvesting system’s sensitivity to small deviations in geometry, material properties, and excitation. In practice, such uncertainties can arise from design tolerances, variability in magnetic material coercivity, or irregular vibration inputs. Since EM harvesters couple magnetic, mechanical, and electrical domains, even minor parameter changes may drastically alter potential energy landscapes, shifting the system between monostable, bistable, and chaotic states. These phenomena fall under uncertainty amplification, which remains insufficiently explored despite its strong implications for industry.

The problem of uncertainty amplification extends beyond individual devices. In future WSN industrial deployments, many energy harvesters could operate in parallel. Lack of reproducibility among these units can compromise the reliability of entire sensor networks, where predictable power delivery is crucial. Therefore, understanding how parameter uncertainties propagate through nonlinear electromagnetic energy harvesting systems is necessary for ensuring consistent and stable performance. Although some recent works have examined probabilistic effects in bistable harvesters [30,31], a comprehensive study linking geometric, material, and excitation uncertainties remains missing.

To address this research gap, the presented study investigates how variations in geometry, magnetic material properties, and excitation forces affect the behavior and energy output of tunable nonlinear electromagnetic harvesters. The nonlinear magnetic restoring forces are characterized through numerical magnetic simulations, providing detailed force–displacement relationships for various magnet configurations. These data are incorporated into a one-degree-of-freedom (1-DOF) dynamic model that captures the coupled electromechanical behavior of the system. Numerical analyses are then performed to assess how uncertainties in magnet spacing, vertical asymmetry, and magnetic coercivity influence the dynamic behavior and harvested power. Additionally, the effects of excitation variability—modeled as harmonic vibration with superimposed impulsive disturbances—are explored to simulate realistic industrial conditions.

Ultimately, this work aims to bridge the gap between laboratory prototypes and mass production of field-ready devices by providing quantitative guidelines for uncertainty-aware design and calibration. In contrast to previous studies that focus primarily on nonlinear behavior itself [32], our contribution lies in revealing how geometric tolerances, magnetic material variability, and excitation irregularities interact to shape the operational regime of tunable electromagnetic harvesters. The outcomes not only advance the theoretical understanding of nonlinear electromagnetic energy harvesters but also establish a foundation for scalable, predictable integration into industrial WSN and Internet of Things (IoT) platforms—where consistent and autonomous operation is crucial for the next generation of smart monitoring technologies.

This work proposes an uncertainty-aware strategy for nonlinear electromagnetic vibration energy harvesters. Rather than attempting to eliminate unavoidable geometric, material, and excitation variability, the strategy deliberately manages these uncertainties to ensure reliable activation of the high-energy operational regime. The strategy combines (i) quantification of uncertainty amplification, (ii) post-assembly geometric tuning to compensate for tolerances and material variability, and (iii) transient excitation to deliberately trigger high-energy operation when ambient excitation is insufficient. The effectiveness of this strategy is demonstrated through coupled FEMM–dynamic simulations under realistic tolerance ranges.

## 2. Modeling of Energy Harvesting System

The energy harvesting system under study is a nonlinear electromagnetic vibration energy harvester (EM VEH) with tunable magnetic stiffness, as shown in Figure 1B.

The harvester consists of a vibrating beam with a moving magnet, coupled to a fixed coil for electromagnetic conversion. Nonlinear restoring forces are introduced by repelling permanent magnets placed near the moving element.

Adjusting the magnet gap δ changes the potential landscape:For large gaps, the system behaves as a softening monostable oscillator;For smaller gaps, it transitions into a bistable regime with two potential wells separated by an energy barrier.

The bistable case is particularly relevant, as cross-well oscillations give the high-amplitude “high-orbit” response associated with maximum energy output. However, this regime exists alongside low-energy intra-well or chaotic solutions, making the overall response highly sensitive to system parameters. Additionally, some transversal asymmetry ε is assumed, which can also alter the potential landscape and system dynamics dramatically.

### 2.1. 1-DOF Model of EM Vibration Energy Harvester

The dynamics can be captured by a 1-DOF nonlinear model with magnetic stiffness and electromagnetic damping, see Figure 2, Table 1 and Equations (1) and (2). The electromechanical representation in Figure 2 is based on the complete force–current analogy, which enables a direct correspondence between electrical network elements and mechanical components such as springs, dampers, and inerters [33].

This reduced model captures the essential multiphysics coupling (mechanical–magnetic–electrical) while remaining useful for analyses with a focus on sensitivity and uncertainty.(1)$$m\ddot{y}+d_{m}\dot{y}+ky+F_{NL}(y,\;\delta ,\varepsilon )+C_{EM}i_{e}=F_{EV}$$(2)$$C_{EM}\dot{y}=L_{C}\frac{di_{e}}{dt}+i_{e}R_{C}+i_{e}Z_{e}.$$
where m is the discretized mass which includes material and geometric parameters of the cantilever beam and tip mass with both magnets, $d_{m}$ is experimentally identified mechanical damping, $C_{EM}$ is electromechanical coupling [29]. $R_{c}$ is the resistance of the coil, $L_{c}$ inductance of the coil, and $Z_{e}$ is the impedance of connected electric loads to the electromagnetic converter. There are two dependent variables, deflection of the tip mass of the beam y, and current in the coil $i_{e}$. The following color representation of the subsystems is followed: mechanical—orange/yellow, magnetic—red, electromagnetic—blue, and excitation—green. We note that the excitation frequency f matches the resonant frequency of a linear version of the energy harvester, i.e., with magnets far apart.

The energy harvesting process can, in principle, be performed on any load $Z_{e}$. However, in this paper, we consider the simplest case of a purely resistive load characterized by resistance $R_{e},$ and the harvester is assumed to operate under the optimal load $R_{e,opt}$(3)$${R_{e}=R}_{e,opt}=\frac{C_{EM}^{2}}{d_{m}}+R_{C}=5250\;\Omega$$

And therefore, the power which is harvested by the EM VEH is calculated as(4)$$P_{e}=i_{e}^{2}R_{e}.$$

The electromechanical coupling coefficient $C_{EM}\;$determines how the electrical load is reflected as an effective mechanical impedance, contributing frequency-dependent electromagnetic damping. The mechanical damping $d_{e}$ represents the baseline linear energy dissipation. In nonlinear electromagnetic harvesters, magnetic restoring forces introduce a state-dependent stiffness, which can be interpreted as a nonlinear modification of the effective mechanical impedance. Small geometric or material uncertainties therefore shift not only the potential landscape but also the nonlinear impedance characteristics of the harvester, leading to transitions between monostable, bistable, and chaotic regimes.

Throughout the paper, excitation is assumed to be harmonic with the possibility of an impulse (Section 5); therefore,(5)$$F_{EV}=m\ddot{z}\sin\left(2\pi ft\right)+F_{imp}\left(t\right),$$
where $\ddot{z}$ is the acceleration of the external harmonic force and f is its frequency. The impulse is represented by a short half-sine pulse of duration T≪1/f(≈4 ms) and strength J, initiated at time $T_{0}$. The impulse term can be written as:(6)$$J=\int_{T_{0}}^{T_{0}+T}F_{imp}\left(t\right)dt=\int_{T_{0}}^{T_{0}+T}F_{0}sin\left(\frac{\pi (t-T_{0})}{T}\right)dt,$$

### 2.2. Modeling Magnetic Nonlinearity

To represent the nonlinear restoring forces, the interaction of permanent magnets is modeled numerically using Finite Element Method Magnetics (FEMM 4.2). The problem is treated as magnetostatic, since the magnetic field is time-invariant. In the simulation:Geometry, magnet positions, and polarization directions are defined.The nonlinear force $F_{NL}(y,\delta )$ is evaluated along the deflection axis y for different magnetic gaps δ.Only the y-component of the force (aligned with beam deflection) is considered. The x-component is neglected due to the assumed small oscillation angles.

The full field distribution and corresponding force curves are shown in Figure 3, which also illustrates the FEMM model setup. The resulting force–displacement characteristics are then incorporated into the system equations of motion.

## 3. Uncertainties in Magnetic Position and Geometry

This section evaluates the geometric component of the proposed uncertainty-aware strategy by quantifying how realistic assembly tolerances affect attractor coexistence and harvested power.

### 3.1. Effect of Magnetic Distance on Generated Power

A critical geometric parameter of the nonlinear electromagnetic harvester is the horizontal magnet spacing δ (see Figure 1). In an idealized laboratory setup, δ is controlled precisely. However, in practical assembly, mechanical tolerances could introduce deviations on the order of ±0.1–0.2 mm, depending on machining and mounting accuracy.

Equations (1) and (2) were numerically integrated for various magnet gaps and to evaluate the effect of δ on the system dynamics under harmonic excitation at frequency f=23 Hz and amplitude $\ddot{z}=0.45\;m/s^{2}$, representative of typical industrial vibration conditions.

Figure 4 shows the corresponding bifurcation diagram of displacement and output power as functions of δ. The results are presented for moderate excitation, where three attractors coexist—a low-energy intra-well (low-orbit) motion, a high-energy cross-well (high-orbit) response, and an intermediate chaotic regime. Simulations were performed for multiple initial displacements to identify these coexisting attractors and to map their domains of attraction. The results highlight that the system behavior is most sensitive near the boundaries between attractors, where small geometric deviations or perturbations can cause abrupt transitions between low- and high-energy responses.

### 3.2. Effect of Magnetic Asymmetry on Power Generation

The role of vertical asymmetry ε in the placement of the nonlinear magnet was systematically investigated for both monostable and bistable configurations of the electromagnetic vibration energy harvester. Figure 5 illustrates the definition of asymmetry ε and the coordinate system used in the analysis.

The results can be summarized as follows:

Monostable configuration—Asymmetry in the potential has a negligible and predictable effect.

Bistable configuration:Low excitation—The response remains trapped inside one potential well regardless of asymmetry. The harvested power is minimal, and the system is poorly utilized.Moderate excitation (Figure 6)—In a perfectly symmetric system, the energy barrier prevents transitions to the high orbit, and the chaotic behavior is the only possible stable solution. An introduced asymmetry lowers this barrier on one side (Figure 7A), enabling cross-well oscillations and improving performance. The high orbit is activated for asymmetry values as one of the possible solutions and occurs only sometimes based on initial conditions. For higher asymmetry values, the chaotic solution no longer exists, and the mean of generated power from coexisting low orbit and high orbit solutions (see Figure 7B) is bigger than that of the original symmetric chaotic solution. Therefore, controlled asymmetry could, for moderate excitation values, activate the otherwise unreachable high orbits.High excitation—The symmetric system consistently reaches the high orbit. In this case, asymmetry is detrimental because it favors one potential well and increases the chance of the response becoming trapped in a single well, reducing harvested power.

### 3.3. Uncertainties in Dimensions of Magnets

Manufacturing tolerances of rare earth magnets are typically around ±0.1 mm, according to their datasheet. Although such deviations appear negligible, they directly affect the total mass of the vibrating beam. Our results show that even small mass changes can significantly influence the dynamic response and harvested power.

This highlights the need to account for real mass values during both modeling and assembly. Fortunately, such deviations can be compensated for by mass tuning, for example, by adding or removing small additional weights during assembly. In this way, the unfavorable impact of dimensional tolerances can be minimized, ensuring consistent energy harvesting performance across devices.

## 4. Uncertainties in Magnetic Material Properties—Coercivity

This section addresses the material-uncertainty component of the proposed strategy, focusing on how coercivity variability alters the magnetic potential and operational regime.

### 4.1. Definition of Materials and Variability Sources

Beyond geometry, material uncertainties of the magnets represent another critical sensitivity factor. Magnets of the same grade (e.g., NdFeB N42) and the same dimensions often show small but non-negligible differences in coercivity due to material inconsistencies or magnetization defects.

In particular, coercivity $H_{c}$—a material property defining resistance to demagnetization—is a critical parameter. For a magnet with dimensions 6×6×20 mm which is typically used in these VEHs, its magnetic properties are defined with maximum energy product ${\left(BH\right)}_{max}$ = $318–350\;kJ/m^{3}$.

Coercivity was estimated from the maximum energy product using the standard analytical relation for hard magnetic materials:(7)$$H_{c}=2\sqrt{\frac{{\left(BH\right)}_{max}}{\mu_{0}}}$$
which represents an upper-bound estimate assuming a linear demagnetization curve. $\mu_{0}$ is the magnetic permeability of free space. For real rare-earth magnets, deviations from linearity and finite recoil permeability typically result in slightly lower effective coercivity. Accounting for this non-ideality, the coercivity corresponding to ${\left(BH\right)}_{max}=318–350\;{\mathrm{kJ}/m}^{3}$ was taken as$${\;H}_{c}\in \left(982,\;1030\right)\;kA/m$$

This seemingly small variation represents a critical source of uncertainty, as it directly influences the shape and depth of the magnetic potential well.

### 4.2. Effect of Coercivity Uncertainty on Potential Landscape

Using the magnetic force data (Section 2.2) for the two boundary values of coercivity, we compute the magnetic potential energy by numerical integration, since:(8)$$F_{NL}=\frac{dU_{NL}}{dy}$$

This is combined with the linear elastic energy $U=ky^{2}/2$ to obtain the total potential energy landscape, which governs the harvester’s dynamic behavior. The simulation results are shown in Figure 8.

It is particularly important to pay close attention to magnet gaps near the boundary between the bistable and monostable regimes, since small parameter changes can cause abrupt transitions in system behavior. This effect is clearly illustrated for the representative magnetic gap of δ = 4 mm, see Figure 9. Suppose one harvester is precisely tuned to operate at δ = 4 mm like a monostable oscillator using a magnet with coercivity $H_{c}=982\;\mathrm{kA}/m$. Another nominally identical harvester, assembled with magnets of slightly higher coercivity $H_{c}=1030\;\mathrm{kA}/m$, will exhibit a distinctly different bistable potential landscape, and its power generation may be drastically affected.

The findings here reinforce the broader message introduced in previous chapters: nonlinear electromagnetic VEH systems are highly sensitive to parameter variation, and uncertainty in magnetic properties can dominate system behavior. To ensure a robust design:Coercivity should be measured or guaranteed within narrow bounds before magnet integration.Fine-tuning of magnet spacing after assembly can compensate for material variability, ensuring consistent operation across devices.

## 5. Uncertainties in Loads

This section evaluates the activation component of the proposed strategy, demonstrating how excitation transients and load variability affect access to high-energy operation.

### 5.1. Excitation Impulses as a High-Orbit Operation Tool

The previous chapter (Section 3) showed that small asymmetries or parameter shifts can help a nonlinear electromagnetic harvester reach high-amplitude orbits under moderate excitation. When the excitation is too weak, however, even controlled asymmetry cannot trigger cross-well motion. Another possible mechanism involves uncertainty of excitation, i.e., short impulsive events capable of activating the high-energy regime.

Industrial vibration signals typically exhibit a dominant periodic component accompanied by occasional transient disturbances. Therefore, the external excitation can be reasonably modeled as a primary harmonic component superimposed with an impulse term that accounts for either stochastic disturbances or deliberate triggering events. Such impulsive excitations may arise from gear tooth impacts, sudden load changes, electrical commutations, or other irregular machine operations.

In this chapter, we specifically examine the case where the harvester initially operates in the low-energy orbit, and the goal is to determine whether impulsive disturbances can transition the system into the high-energy orbit. To evaluate the impact of impulsive disturbances, the EM VEH was simulated under varying impulse onset times and amplitudes, as formulated in Equations (5) and (6). The resulting responses are illustrated in Figure 10. Numerical simulations show that:The system response repeats every excitation period.There exists an optimal impulse strength maximizing the chance of reaching the high orbit.Too weak impulses leave the motion trapped; too strong ones overexcite the system, and its behavior is more unpredictable.Large impulses affect the generated power even if the system does not make the jump.

Further simulations for various harmonic amplitudes $F_{0}$ (see Figure 11) reveal that impulsive excitations are most beneficial in the moderate-excitation regime—strong enough to overcome the potential barrier, yet below the threshold where the harmonic force alone ensures high-orbit motion. Figure 11 provides a map of impulse timing and strength combinations that enable a transition to the high-energy orbit, as well as the conditions that result in no orbit transition.

From a practical standpoint, such a map can be used during design or device calibration to determine the minimum impulse required to reliably activate the high-energy orbit under given operating conditions—for example, identifying whether naturally occurring industrial shocks are sufficient, or whether an intentional control impulse must be applied to maintain high-output operation.

From an energy point of view, the half-sine impulse acts as an investment: we deliberately spend an amount of energy $E_{\mathrm{imp}}$ to drive the system from the low-orbit to the high-orbit regime.

From an energetic perspective, the applied half-sine impulse can be interpreted as an intentional energy investment used to drive the system from a low-energy orbit into the high-energy (cross-well) regime. The impulse adds momentum to the oscillating mass and increases its kinetic energy. The minimum kinetic energy associated with an impulse J applied to a system with effective mass m can be approximated as(9)$$E_{imp}=\frac{J^{2}}{2m}$$

For the investigated harvester (m=90 g) and a representative impulse strength J=0.08 Ns, the injected energy is approximately $E_{imp}\approx \;0.036\;J$. This value represents a lower bound, neglecting losses during the short impulse duration, but provides a useful order-of-magnitude estimate of the energy required to activate high-orbit motion.

It should be noted that impulsive excitation does not universally promote transitions to higher-energy responses. The effect of an impulse depends strongly on its timing relative to the system phase and on the current attractor. While properly timed impulses can drive the system from low-energy or chaotic motion into the high-orbit regime, unfavorable timing may instead cause a transition from a high-energy or chaotic response into a low-energy orbit. Therefore, reliable upward transitions require either favorable natural timing or knowledge of the current operating state, allowing impulses to be applied selectively and possibly repeatedly.

### 5.2. Uncertainties in Electrical Load

The analyzed EM VEH could be used to power the data transmission of an autonomous sensor node that measures strain on a structure under vibration, as a similar lab self-power wireless sensor was developed and tested in our lab [34]. For such a self-monitoring wireless task, an average of 7 mW of power is required for its continuous operation, with a power consumption of 4 mW for sensing and a power consumption of 12 mW for transmission of data. Advanced control of sensing, transmitting, and idle operation modes has a significant impact on the coupled electromagnetic force in Equation (1). In the case of dynamic responses of a nonlinear energy harvesting system, the advanced control of current by power consumption could be in the form of active excitation impulses, as was presented in the previous subsection. It is evident that control of output power consumption can significantly affect the coupled electromagnetic force, which, in the form of variable feedback force, affects the solution, as shown in Figure 10. An active approach to managing electricity consumption and influencing the solution of a nonlinear energy harvesting system deserves deeper analysis in a new research paper.

Interestingly, the irregular impacts present in industrial excitation spectra—traditionally seen as noise—need not be detrimental. It is clear from the results of this chapter that random impulses can help the harvester jump from low- to high-energy orbits. Two approaches emerge:Passive use—design the harvester so it can naturally respond to random shocks, letting these occasional disturbances push it into the high-energy regime.Active use—apply controlled electrical impulses when the system operates in low-energy states.

Understanding uncertainties in both excitation and electrical load, therefore, complements geometric and material analyses, offering a clear path toward robust and adaptable nonlinear electromagnetic harvesters.

## 6. Discussions

Uncertainties play a defining role in the behavior and industrial deployment of nonlinear electromagnetic vibration energy harvesters. Taken together, the results demonstrate that the proposed uncertainty-aware strategy reliably enables and maintains high-energy operation despite realistic geometric, material, and excitation uncertainties. The harvester exhibits optimal performance when operating in the high-amplitude, or high-orbit, regime, where energy conversion is maximized. Achieving and maintaining this effective regime requires careful consideration of:Magnetic material properties;Horizontal magnet position (δ);Vertical magnet position (ε).

The results show that even small variations in these parameters—well within typical manufacturing tolerances—can lead to large differences in dynamic response. Consequently, two harvesters with identical nominal specifications may exhibit entirely different operational regimes: one functioning efficiently in the desired nonlinear orbit, while another remains confined to a low-energy state.

At the same time, the same mechanisms that introduce uncertainty can also be leveraged for performance tuning.

Controlled asymmetry or transient impulses can assist transitions into the high-energy orbit under weak or moderate excitation.Adjustable magnet positioning in longitudinal and transverse directions enables post-assembly fine-tuning of individual devices.

It is therefore clear (from Figure 4 and Figure 6) that the high orbit is capable of generating the required power for the continuous operation of industrial wireless sensor nodes. However, the chaotic or low-orbit behavior is not capable of harvesting the required outputs. The active approach for controlling power consumption in WSN should also be a way to transform chaotic or low-orbit behavior into a high-orbit solution with sufficient power output.

The energetic cost of impulsive excitation must be considered when evaluating its practical applicability. Although the impulse represents an external energy input, its role is limited to initiating a transition between coexisting attractors rather than sustaining the system response.

Once the high-orbit regime is activated, the average harvested power increases by several milliwatts compared to low-orbit or chaotic motion. Consequently, the energy invested by the impulse is recovered within a short time span (on the order of seconds), after which the harvester produces a net positive energy gain. Future work could focus on automated calibration and self-tuning mechanisms, including state-aware impulsive triggering strategies.

From a practical perspective, the presented uncertainty analysis provides a direct basis for fine-tuning electromagnetic energy harvesters prior to field deployment. The identified sensitivity to geometric tolerances, magnetic coercivity, and excitation variability enables several concrete calibration strategies: (i) post-assembly adjustment of magnet spacing and transverse alignment to compensate for manufacturing tolerances, (ii) selection or sorting of magnets based on measured coercivity to reduce unit-to-unit variability, and (iii) controlled application of mechanical or electrical impulses during commissioning to deliberately trigger and stabilize high-orbit operation. Such procedures will be essential for industrial applications, particularly in railway environments, where devices must operate consistently under variable and partially unpredictable vibration conditions, such as railway infrastructure monitoring.

The presented uncertainty-aware strategy directly builds on previously validated railway energy harvesting systems [28,29,35], extending their applicability by addressing the sensitivity and reproducibility challenges that arise when moving from linear prototypes toward tunable nonlinear harvesters for real railway environments.

## 7. Conclusions

From a design perspective, mechanical adjustability and wide basins of attraction are essential for robust operation of industrial energy harvesting systems for WSNs. This study demonstrates that the key challenge—and the core novelty of our contribution—is not the creation of the nonlinearity but understanding how unavoidable geometric, material, and excitation uncertainties amplify through the nonlinear system and determine whether a device is usable or not. Parameter deviations and uncertainties cannot be eliminated, but they can be quantified, understood, and harnessed—allowing each harvester to be calibrated for optimal performance in its specific deployment environment.

In industrial mass production, such tuning can be achieved, for example, by fine-adjusting magnet spacing during final assembly or maintenance, much like the calibration of sensors or mass-balancing of rotating machinery. Implementing simple mechanical adjustment screws would enable in situ optimization and consistent energy output despite small geometric deviations or changing vibration sources.

The lack of reproducibility represents a critical challenge for small-series industrial production and integration into larger WSN or IoT frameworks, where consistent and predictable performance across devices is essential. This work demonstrates that the reliability of large-scale autonomous sensing systems depends not only on the performance of individual nonlinear electromagnetic harvesters but also on their reproducibility across multiple units. The sensitivity of bistable and tunable magnetic systems to small parameter deviations highlights the need for design concepts that ensure repeatable operation across devices produced within standard manufacturing tolerances.

In addition to geometric and material tuning, the results demonstrate that impulsive excitation represents an effective and energetically justified mechanism for activating high-orbit operation in nonlinear electromagnetic harvesters. Applying short mechanical or electrical impulses could be justified since they can be used to deliberately drive the system from low-energy or chaotic responses into the high-energy regime, where sustained power output is significantly higher. The impulse is required only to initiate the transition rather than sustain the response. Impact excitation can be viewed as a practical tool for commissioning, calibration, or occasional reactivation of harvesters operating under weak or variable ambient excitation.

By combining sensitivity analysis, parameter control, and post-assembly tuning, reproducible performance can be achieved even in small-series production. These strategies support scalable industrial deployment of nonlinear vibration energy harvesting technology for autonomous sensing and predictive maintenance in the railway and aviation industries.

The key contribution of this work is showing that reproducible performance of nonlinear electromagnetic energy harvesters can be achieved through an uncertainty-aware strategy that combines sensitivity quantification, geometric tuning, and controlled activation.

Future work will focus on automated calibration and self-tuning mechanisms that compensate for magnetic and geometric tolerances, enabling consistent plug-and-play integration of energy harvesters into industrial WSN and IoT systems.

## Figures and Tables

**Figure 1 sensors-26-00253-f001:**
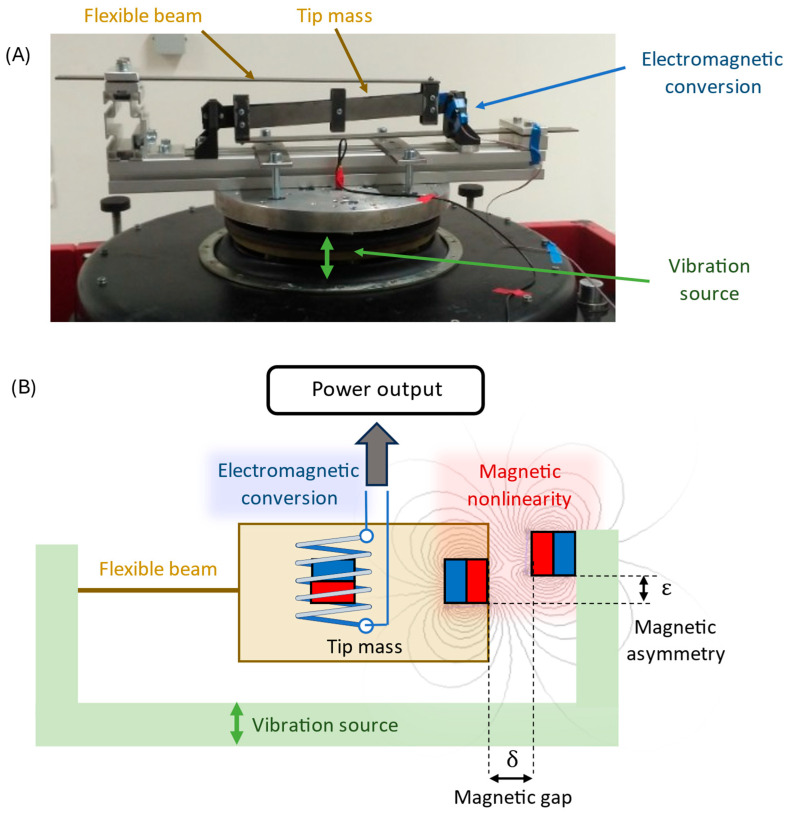
(**A**) Experimental setup of the hybrid electromagnetic energy harvester mounted on an electrodynamic shaker [29]. (**B**) Schematic representation of the operating principle highlighting electromagnetic energy conversion, magnetic nonlinearity, and intentional magnetic asymmetry defined by the gap δ and offset ε.

**Figure 2 sensors-26-00253-f002:**
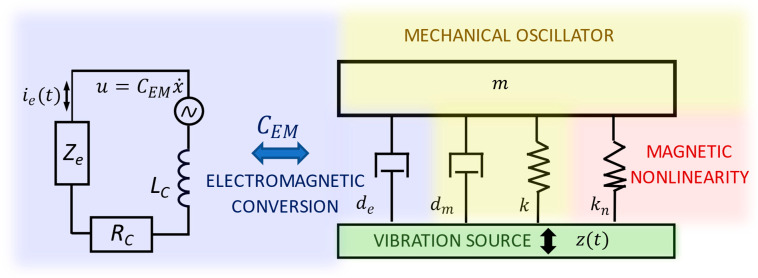
Electromechanical model of the nonlinear electromagnetic vibration energy harvester showing the interaction between the mechanical oscillator with magnetic nonlinearity and the electrical conversion circuit. The electromagnetic coupling coefficient $C_{EM}$ mediates energy transfer between the vibrating mass and the electrical subsystem. Color shading indicates system domains: yellow—mechanical, blue—electromagnetic, green—excitation source, red—magnetic nonlinearity.

**Figure 3 sensors-26-00253-f003:**
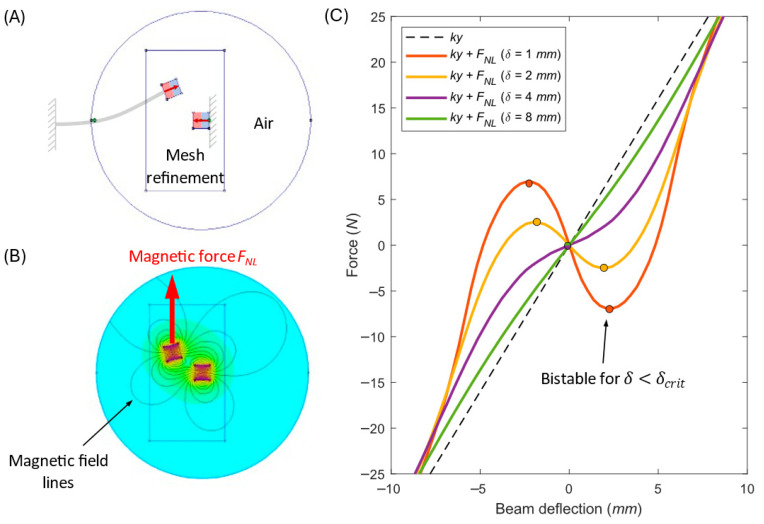
FEMM magnetostatic model of the nonlinear magnetic force. (**A**) Geometry, boundary, and mesh-refined region. (**B**) Resulting magnetic flux density distribution B, which is recalculated into the magnetic force $F_{\mathrm{NL}}$. (**C**) Total restoring force $F_{\mathrm{tot}}\left(y\right)=k\;y+F_{\mathrm{NL}}\left(y,\delta \right)\;$ for several gaps δ. Filled markers: stable equilibria $\left(dF_{\mathrm{tot}}/dy>0\right)$. Dashed black: linear stiffness k y.

**Figure 4 sensors-26-00253-f004:**
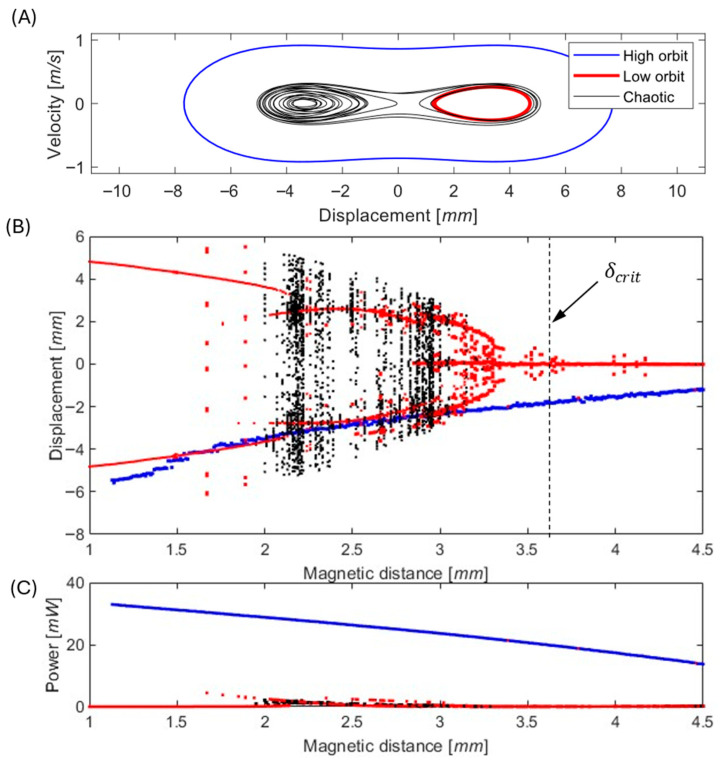
Influence of magnetic distance δ on the nonlinear dynamics and harvested power of the electromagnetic vibration energy harvester. (**A**) Phase portraits (velocity–displacement) illustrating coexisting dynamic responses at δ=2.2 mm for various initial conditions. (**B**) Bifurcation diagram of displacement as a function of magnetic distance, highlighting the transition between monostable and bistable regimes; the critical distance $\delta_{crit}=3.6\;$ mm is indicated. Corresponding average harvested power versus magnetic distance. (**C**) Corresponding average harvested power versus magnetic distance. High-orbit solutions are shown in blue, low-orbit solutions in red, and chaotic responses in black.

**Figure 5 sensors-26-00253-f005:**
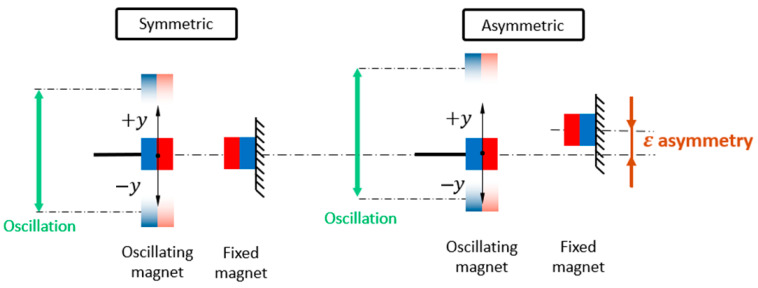
Schematic of symmetric and asymmetric nonlinear magnetic energy harvester configurations. In the asymmetric case, the fixed magnets are vertically shifted by ε, introducing potential misalignment while the oscillation zero remains unchanged.

**Figure 6 sensors-26-00253-f006:**
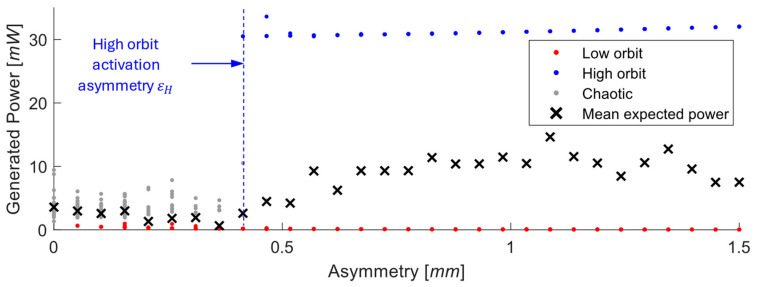
Effect of asymmetry on power generation of the EM VEH. $\varepsilon_{H}=0.4\;mm$ denotes the minimum asymmetry that makes reaching the high orbit possible. Simulations performed for $\ddot{z}=10\;m/s^{2}$.

**Figure 7 sensors-26-00253-f007:**
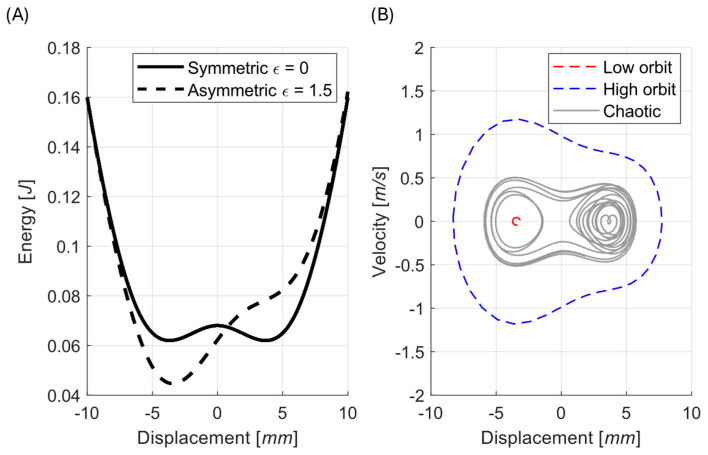
Example magnetic asymmetry ε=1.5 mm and its effect on the potential landscape (**A**) and possible solutions (**B**). The only chaotic solution for the symmetric case (gray solid line) is destroyed by the asymmetry, and low-orbit (red) and high-orbit (blue) solutions now coexist (dashed lines). Simulations performed for $\ddot{z}=10\;m/s^{2}$.

**Figure 8 sensors-26-00253-f008:**
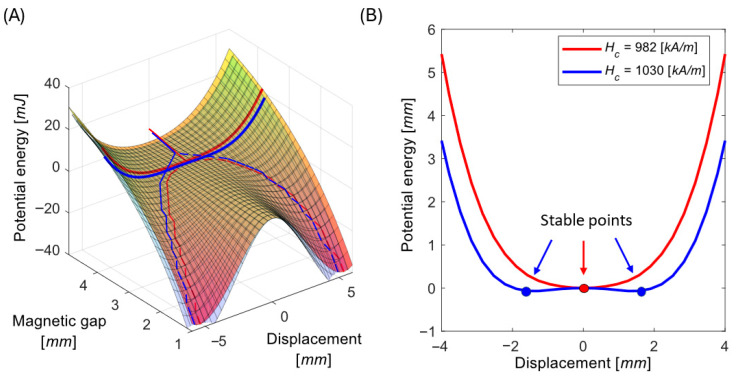
Potential energy landscape of the energy harvesting system as a combined function of displacement and magnetic gap for lower (982, red shades) and higher (1030, blue shades) limit of coercivity (**A**). Dashed lines show the position of fixed oscillation points. Thick solid lines correspond to lines in plot (**B**) which are a section of the potential for sample magnetic gap δ=4 mm.

**Figure 9 sensors-26-00253-f009:**
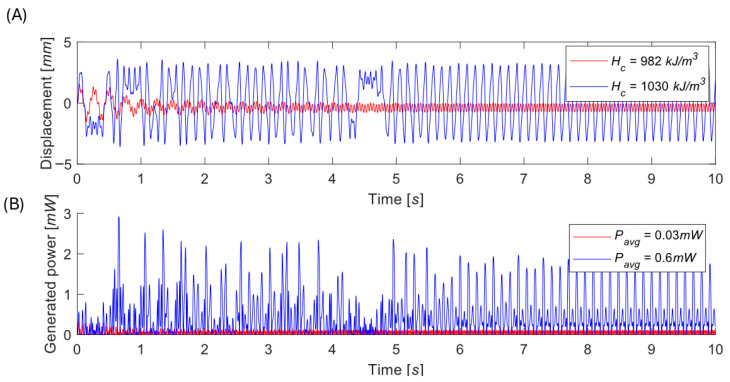
Comparison of the behavior of two energy harvesters built with magnets of identical size and grade, which exhibit different behavior due to property variations within manufacturer-specified tolerances. Time evolution of oscillations (**A**) and generated power (**B**). Simulations performed for $\ddot{z}=6.5\;m/s^{2}$.

**Figure 10 sensors-26-00253-f010:**
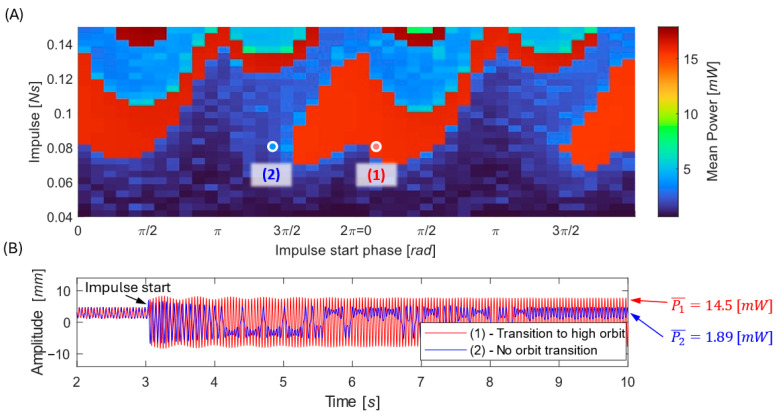
Generated power as a combined function of impulse starting time and impulse strength (**A**). Time histories of the oscillation (**B**) showing cases where the harvester—initially in the low-energy orbit—does transition to the high-energy orbit after an impulse (red) and cases where it does not transition (blue). Simulations performed for $\ddot{z}=4.5\;m/s^{2}$.

**Figure 11 sensors-26-00253-f011:**
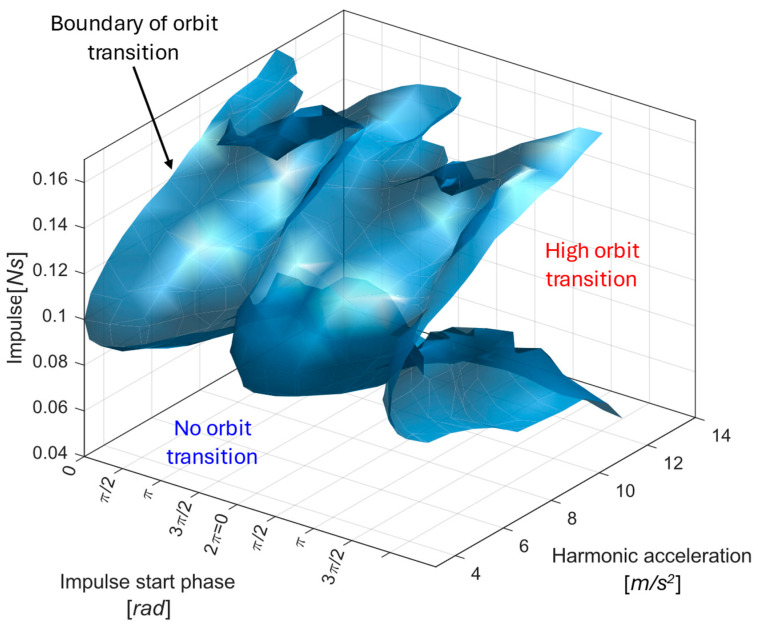
Parameter map illustrating impulse-induced orbit transitions in the electromagnetic vibration energy harvester. The surface represents the orbit-transition boundary (separatrix) in the parameter space defined by impulse magnitude, impulse start phase, and harmonic base-acceleration amplitude. This boundary separates combinations of impulse excitation that trigger a transition from the low-energy orbit to the high-energy orbit from those that do not result in an orbit jump. Regions corresponding to successful high-orbit transitions and no-transition responses are indicated.

**Table 1 sensors-26-00253-t001:** Nominal mechanical, electromagnetic, and excitation parameters of the electromagnetic vibration energy harvester.

System	Parameter	Symbol	Nominal Value	Unit
Mechanical	Mass	m	90	g
	Damping (mechanical)	$d_{m}$	0.034	Ns/m
	Stiffness	k	3178	N/m
Electromagnetic	EM coupling	$C_{EM}$	13.16	N/A
	EM load resistance	$Z_{e}$	5250	Ω
	Coil resistance	$R_{C}$	149	Ω
	Coil inductance	$L_{C}$	neglected	mH
Excitation	Excitation amplitude	$\ddot{z}$	range 2–25	$m/s^{2}$
	Excitation frequency	f	23	Hz

## Data Availability

The data that support the findings of this study are openly available in Zenodo. Dataset link: https://doi.org/10.5281/zenodo.17565040.

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
