# Peer review of "Influence of Geometric and Material Uncertainties on the Behavior of Monostable and Bistable Electromagnetic Energy Harvesters"

_sensors, 2025, doi:10.3390/s26010253_

Round 1

Reviewer 1 Report

Comments and Suggestions for Authors

This is an interesting paper, highlighting how an electromagnetic energy harvester can be tuned to various response regimes and made to optimise its power output. Moreover, the paper gives detailed analysis of how various sources of uncertainty can make a dramatic difference to the performance of the isolator. A particularly interesting aspect is the argument that occasional impulses, which may be present in the excitation naturally, could be exploited to help promote the resonant responses that give the best power output, and perhaps further detail on this would be beneficial. The paper is well written and with a good literature survey, and I believe can be accepted subject to some minor comments below.

  1. Fig10b, what is the meaning of ‘transfer’ or ‘no transfer’ in the legend?
  2. The discussion of potential impulses is interesting but raises a few questions:
    1. Is it possible to quantify the energy given by the impulse compared to the additional energy recovered? This would answer weather a mechanism for deliberately impacting the system to trigger resonance, expending energy to gain greater energy , could be justified.
    2. Is there a situation in which an impulse response is more likely to trigger a jump up rather than a jump down?
    3. The conclusions don’t seem to follow up on any of this – something on impact excitation should be added.
  3. (not essential) the authors could consider the following paper to include in the literature review, as it concerns a tuneable device in response to similar issues to those in this paper:

Madinei, H., Khodaparast, H., Adhikari, S. et al. Adaptive tuned piezoelectric MEMS vibration energy harvester using an electrostatic device. Eur. Phys. J. Spec. Top. 224, 2703–2717 (2015). https://doi.org/10.1140/epjst/e2015-02584-6

Author Response

Comment 1: Fig10b, what is the meaning of ‘transfer’ or ‘no transfer’ in the legend?

Response 1: We thank the reviewer for pointing out this ambiguity. In Fig. 10b, the two labels indicate whether the impulsive disturbance is sufficiently strong and correctly timed to move the nonlinear energy harvester from the unfavourable low-energy orbit into the favourable high-energy orbit. In other words, “transfer” indicates that the impulse triggers a transition to the high-energy orbit, whereas “no transfer” indicates that the system remains in the low-energy orbit. To improve clarity, we updated the description in both the figure legend and figure caption to “transition to high orbit” and “no orbit transition.” (Page 12, lines 336-339)

Comment 2: The discussion of potential impulses is interesting but raises a few questions:

Comment 2a: Is it possible to quantify the energy given by the impulse compared to the additional energy recovered? This would answer weather a mechanism for deliberately impacting the system to trigger resonance, expending energy to gain greater energy, could be justified.

Response 2a: Yes, it is. The energy associated with the applied impulse can be quantified and compared with the additional harvested energy. The minimum kinetic energy injected by an impulse is . For the investigated harvester (  kg) and a representative impulse N·s, this corresponds to approximately 0.036 J. Numerical results show that once such an impulse triggers a transition to the high-energy orbit, the average harvested power increases by several milliwatts, allowing the invested energy to be recovered within a few seconds of operation. Since the impulse is required only to initiate the transition and not to sustain the response, deliberate impulsive triggering is energetically justified. The sections were revised, and an energy analysis was added to the paper. (Page 13, lines 362-371 and Page 15, lines 434-443)

Comment 2b: Is there a situation in which an impulse response is more likely to trigger a jump up rather than a jump down?

Response 2b: Yes, there is. An impulse is more likely to trigger an upward transition when the harvester operates in a low-energy or chaotic orbit and the impulse is applied at a favorable phase of the oscillation. Numerical results show that impulse timing strongly affects the outcome: properly timed impulses can promote transitions to the high-energy orbit, whereas poorly timed impulses may instead drive the system into a lower-energy or trapped state. In practice, reliable upward transitions would require knowledge of the current operating orbit and phase, enabling impulses to be applied selectively; otherwise, additional corrective impulses may be needed. This section was revised and explained more clearly. (Page 14, lines 372-379)

Comment 2c: The conclusions don’t seem to follow up on any of this – something on impact excitation should be added.

Response 2c: We agree. The chapter Conclusions has been extended to explicitly address impact (impulsive) excitation. The revised text now highlights impulsive excitation as an energetically justified and practical mechanism for activating high-orbit operation, emphasizing its role in commissioning, calibration, and reactivation of nonlinear electromagnetic harvesters under weak or variable excitation conditions. (Page 16, lines 477-485)

Comment 3: (not essential) the authors could consider the following paper to include in the literature review, as it concerns a tuneable device in response to similar issues to those in this paper: 

Madinei, H., Khodaparast, H., Adhikari, S. et al. Adaptive tuned piezoelectric MEMS vibration energy harvester using an electrostatic device. Eur. Phys. J. Spec. Top. 224, 2703–2717 (2015). https://doi.org/10.1140/epjst/e2015-02584-6

Response 3: We thank the reviewer for this helpful suggestion. The recommended paper has now been incorporated into the revised literature review. In particular, we highlight it as a notable example of a hybrid architecture in which only one transduction mechanism (the piezoelectric subsystem) is used for energy harvesting, while the electrostatic subsystem is employed exclusively for tuning the resonance frequency. This distinction aligns well with our discussion on tunable systems and supports our motivation for examining parameter sensitivity in nonlinear, adaptable energy harvesters. The corresponding sentence has been added in the Introduction. (Page 2, lines 52-57)

Reviewer 2 Report

Comments and Suggestions for Authors

This paper investigates how variations in geometry, magnetic material properties, and excitation forces affect the behavior and energy. Some comments should be considered:
1. Presentation of figures needs to be improved. For example, It is recommended to add some tags in Fig.1a to show more details about the EM VEH component.
2. The readability of Figure 11 is not good, Please rephrase it.
3. Regarding the Electromechanical analogy in figure 2, some references are absent(A review of the inerter and inerter-based vibration isolation: theory, devices, and applications. J Frankl Inst-Eng Appl Math. 2022;359(14):7677–7707. doi:10.1016/j.jfranklin.2022.07.030). 
4.  It is recommended to incorporate references from the past three years to enhance the innovative aspects of this paper.
5. Regarding Figure 9 and other areas, the unit writing style should be checked carefully( regular OR italic).
6. It usually uses magnetic force to realize a muti-stable condition, so what is the Innovation of this research?
7. Some applications or providing methods for solving practical problems are suggested to be added in the perspective works in Section 6.

Author Response

Comment 1: Presentation of figures needs to be improved. For example, It is recommended to add some tags in Fig.1a to show more details about the EM VEH component.

Response 1: Thank you for the suggestion. We have updated Figure 1(A) to include clear annotations of the EM VEH components, consistent with the style used in Figure 1(B). In addition, all figures throughout the manuscript have been revised to improve readability, including increased font sizes and clearer labeling. (Page 4, line 130).

Comment 2: The readability of Figure 11 is not good, Please rephrase it.

Response 2: Thank you for this helpful comment. We have clarified in both the main text and the figure captions that Figures 10 and 11 specifically analyze the case where the harvester initially operates in the low-energy orbit and the objective is to determine whether an impulse can transition the system into the high-energy orbit (Page 12, lines 322-324). The captions now explicitly state this assumption, and the related text in Section 5.1 has been revised accordingly to improve clarity. We believe these changes significantly enhance the interpretability of both figures. (Page 13, lines 346-358).

Comment 3: Regarding the Electromechanical analogy in figure 2, some references are absent(A review of the inerter and inerter-based vibration isolation: theory, devices, and applications. J Frankl Inst-Eng Appl Math. 2022;359(14):7677–7707. doi:10.1016/j.jfranklin.2022.07.030).

Response 3: We thank the reviewer for the suggestion. The referenced paper provides a comprehensive review of inerter-based vibration isolation and mechanical network synthesis. However, our work does not employ inerters or inertance-based elements, nor do these concepts appear in the modeling, analysis, or electromechanical coupling illustrated in Figure 2. Because the inerter framework is not relevant to the electromagnetic harvester architecture studied here, we believe that including this reference would be outside the scope of the paper. For this reason, we have not added the citation.

Comment 4: It is recommended to incorporate references from the past three years to enhance the innovative aspects of this paper.

Response 4: Thank you for the suggestion. We have revised the manuscript and incorporated additional relevant references [3], [5], [6], [8], [12] and [32] from the past three years, particularly in the Introduction, to better reflect recent advances and to further highlight the innovative aspects of the present work. (Page 1, lines 33, 37, Page 2, lines 41, 44, Page 3, line 107)

Comment 5: Regarding Figure 9 and other areas, the unit writing style should be checked carefully( regular OR italic).

Response 5: Thank you for this observation. We have carefully reviewed the entire manuscript, including all figures, and unified the unit formatting throughout to ensure consistency in writing style.

Comment 6: It usually uses magnetic force to realize a muti-stable condition, so what is the Innovation of this research?

Response 6: Thank you for the question. The innovation of our work is not in introducing magnetic multistability itself, but in analyzing how geometric, material, and excitation uncertainties affect transitions between monostable, bistable, and chaotic regimes, and how these uncertainties can be managed or exploited to ensure robust and reproducible operation of nonlinear electromagnetic harvesters.

Based on the reviewer’s comment, we have revised the Introduction (Page 3, lines 106-109) and Conclusions (Page 15, lines 457-461) to make the novelty of this uncertainty-focused framework more explicit and easier for readers to identify. The updated text now clearly highlights that our contribution addresses a critical gap in understanding and controlling uncertainty amplification in tunable nonlinear EM harvesters, which has not been covered in previous studies.

Comment 7: Some applications or providing methods for solving practical problems are suggested to be added in the perspective works in Section 6.

Response 7: Thank you for this valuable suggestion. We have extended Section 6 to include concrete methods for translating the presented analysis into practical solutions. The revised text now outlines specific approaches applied during commissioning to ensure reliable high-orbit operation. These methods are discussed in the context of industrial applications for railway systems. (Page 15, lines 444-454)

Round 2

Reviewer 2 Report

Comments and Suggestions for Authors

Thanks for the authors' effort to reply to the queries or comments. The revised paper still cannot be considered for publication since the corresponding response does not stick to the point of the comments. Concerns continue to grow:
1. Presentation of figures needs to be improved.
2. The analysis of the proposed strategy still does not bring out the contribution effectively.
3. Regarding Comment 3, the Electromechanical analogy in figure 2 still needs the reference. 
4.  Regarding Comment 6, some approach for nonlinear dynamics equivalent impedance analysis of electromagnetic harvesters is recommended to be discussed.
5.  Application in railway environments is a promising area, are there some sucessful application?
6. Regarding the Data Availability Statement, the website address seem not to be a activation link.  

Author Response

Comment 1: Presentation of figures needs to be improved.

Response 1: Thank you for the comment, we identified the figures and tables that required the most improvement in terms of presentation and clarity, and revised them accordingly. Specifically, Figure 1 was improved with better color consistency, clarity, and an updated caption; Figure 4 was simplified by consolidating multiple subplots into a clearer representation of system behaviors; and Figure 11 and its caption were revised again to improve the interpretability of the parameter map. In addition, Table 1 was restructured to enhance readability and notation consistency.

Comment 2: The analysis of the proposed strategy still does not bring out the contribution effectively.

Response 2: We thank the reviewer for this comment. To clarify the contribution, we explicitly defined the proposed uncertainty-aware strategy in the Introduction (Page 3, lines 114-122) and consistently linked the analyses in Sections 3–5 (Page 7, lines 209-211; Page 10, lines 288-290 and Page 12, lines 338-340) to its geometric, material, and activation components. The Discussion (Page 16, lines 444-446) and Conclusions (Page 18, lines 532-535) were also revised to explicitly summarize how these results demonstrate the effectiveness of the proposed strategy under realistic uncertainty conditions.

Comment 3: Regarding Comment 3, the Electromechanical analogy in figure 2 still needs the reference. 

Response 3: We thank the reviewer for this comment. The suggested review paper has now been added to the main text where the electromechanical analogy underlying Figure 2 is introduced. (Page 5, lines 151-154)

Comment 4: Regarding Comment 6, some approach for nonlinear dynamics equivalent impedance analysis of electromagnetic harvesters is recommended to be discussed.

Response 4: Yes, the nonlinear equivalent impedance framework is directly linked to the electromechanical coupling coefficient , which governs load-reflected electromagnetic damping, while mechanical damping forms the linear baseline impedance. In nonlinear magnetic harvesters, uncertainties primarily perturb the state-dependent impedance introduced by magnetic stiffness rather than the coupling itself. Modified on Page 6, lines 177-185.

Comment 5:  Application in railway environments is a promising area, are there some sucessful application?

Response 5: Yes, successful applications of electromagnetic vibration energy harvesting in railway environments have been demonstrated. The manuscript has been revised to explicitly reference validated laboratory and field-tested railway harvesters, including a deployed electromagnetic trackside harvester powering a wireless sensor node and experimentally verified cantilever-based railway harvesters, thereby clarifying the practical relevance of the proposed approach for railway applications. (Page 17, lines 491-494)

Comment 6: Regarding the Data Availability Statement, the website address seem not to be a activation link.  

Response 6: Thank you for pointing this out. The dataset has now been updated and made publicly available. The Data Availability Statement has been corrected, and the link is now active and accessible at: https://doi.org/10.5281/zenodo.17565040
